# Structure, Activity and Function of the Dual Protein Lysine and Protein N-Terminal Methyltransferase METTL13

**DOI:** 10.3390/life11111121

**Published:** 2021-10-21

**Authors:** Magnus E. Jakobsson

**Affiliations:** Department of Immunotechnology, Lund University, Medicon Village, 22100 Lund, Sweden; Magnus.Jakobsson@immun.lth.se

**Keywords:** post translational modification, lysine methylation, N-terminal methylation, translation, enzyme specificity, eEF1A, METTL13

## Abstract

METTL13 (also known as eEF1A-KNMT and FEAT) is a dual methyltransferase reported to target the N-terminus and Lys55 in the eukaryotic translation elongation factor 1 alpha (eEF1A). METTL13-mediated methylation of eEF1A has functional consequences related to translation dynamics and include altered rate of global protein synthesis and translation of specific codons. Aberrant regulation of METTL13 has been linked to several types of cancer but the precise mechanisms are not yet fully understood. In this article, the current literature related to the structure, activity, and function of METTL13 is systematically reviewed and put into context. The links between METTL13 and diseases, mainly different types of cancer, are also summarized. Finally, key challenges and opportunities for METTL13 research are pinpointed in a prospective outlook.

## 1. Introduction

Cellular protein synthesis is guided and catalyzed by the ribosome, which uses messenger RNA as a template for protein synthesis in a process termed translation. Several elongation factors support the process of translation, and one prominent example is the eukaryotic elongation factor 1 alpha (eEF1A), which delivers aminoacyl-tRNA complexes to the ribosome acceptor (A)-site to provide substrate for protein synthesis. The function of a protein is often regulated by enzyme-mediated post-translational modification (PTM) [1]. Prominent examples include phosphorylation, glycosylation, acetylation, and methylation [2].

In cells, specific methyltransferase (MT) enzymes catalyze the transfer of a methyl group (-CH_3_) from *S*-adenosylmethionine (AdoMet) to specific substrates to generate a methylated product and *S*-adenosylhomocysteine (AdoHcy) (Figure 1A,B). Protein methylation has been most extensively studied on lysine [3] and arginine [4], but emerging evidence suggests that also histidine methylation [5,6] is prevalent and important. In addition, methylation can occur on the side chains of glutamate, glutamine, asparagine, and cysteine as well as the protein N-terminus (Nt) and C-terminus [7]. Methylation of lysine and the protein Nt are biochemically similar. They both occur on primary amino groups corresponding to the α-amino group of the protein Nt and the ε-amino group of the lysine side chain (Figure 1C,D). Each amino group can accept up to three methyl groups yielding mono-, di-, and tri-methylated substrates (Figure 1C,D).

The protein Nt α-amino group and the ε-amino group of lysine are both chemical bases and exist in both possible protonation states; a neutral state and a positively charged state. The neutral, i.e., unprotonated state is characterized by a free electron pair capable of acting as a nucleophile in nucleophilic substitution reactions [8]. In cells, the differentially protonated forms are in equilibrium and their relative abundance is determined by the acid dissociation constant (pKa) and pH. Notably, the protein Nt has a pKa close to physiological pH whereas a lysine side chain typically has a pKa above 10 [9]. Consequently, the Nt is more chemically active under physiological conditions.

Methylation of the Nt and lysine side chain have similar biochemical consequences. Firstly, methylation increases both the void occupancy and the hydrophobicity. Secondly, trimethylation renders a permanent positive charge and chemically saturates the amino group making it chemically inert. For reference, both sites can be acetylated by acetyl-CoA dependent acetyltransferases (Figure 1E). Nt and lysine acetylation also renders the amino groups chemically inert by occupying the “free” election pair but, in contrast to methylation, acetylation neutralizes the positive charge (Figure 1F,G).

It was recently reported that human methyltransferase-like protein 13 (METTL13) (also called eEF1A-KNMT or FEAT) trimethylates the Nt and dimethylates a specific lysine in position 55 in eEF1A (eEF1A-Lys55) to regulate mRNA translation and protein synthesis [10,11,12]. Here, we review the literature on METTL13 and discuss structural, biochemical, and cellular features as well as its links to disease. We end with a prospective outlook and propose directions for future research.

## 2. Structural Features

The human genome is predicted to encode over 200 enzymes with MT activity [13]. These are often categorized based on structural features and MT activity has been reported for five distinct protein folds [14]. The largest group of MTs corresponds to the so-called seven beta strand (7BS) domain containing enzymes that harbor a characteristic fold comprising 7BS and alternating alpha helices (Figure 2A).

METTL13 has a unique domain organization comprising two distinct 7BS domains, henceforth denoted MT13-N and MT13-C (Figure 2B–D). Although the domains belong to the same 7BS superfamily, they are not closely related [11,13]. The closest paralog for MT13-C is spermidine synthase, an enzyme that catalyzes the transfer of a propylamine group from *S*-adenosylmethioninamine to putrescine to generate spermidine [13,15] (Figure 2E). In contrast, the MT13-N domain has three close paralogs that are all established lysine-specific MTs, namely to CS-KMT [16,17] (also called METTL12), eEF1A-KMT4 (previously annotated as a splice variant of ECE2) [18] and eEF1A-KMT2 (also called METTL10) [19] (Figure 2E).

The evolutionary conservation of METTL13 has been explored through systematic BLAST searches throughout the eukaryotic kingdom [11]. This analysis revealed clear paralogs in several commonly used model organisms including *D. melanogaster*, *C. elegans*, and *A. thaliana* but not in *S. cerevisiae* [11]. In line with these observations, the *S. cerevisiae* eEF1A homolog lacks methylation at the site corresponding to human eEF1A-Lys55 [20]. However, *S. cerevisiae* eEF1A is Nt trimethylated and the responsible enzyme has been identified as YLR285Wp [21], a 7BS MT that bears no sequence homology to MT13-C [11,13]. This demonstrates that eEF1A Nt MT activity has arisen twice in evolution, underscoring the functional importance of the PTM.

**Figure 2 life-11-01121-f002:**
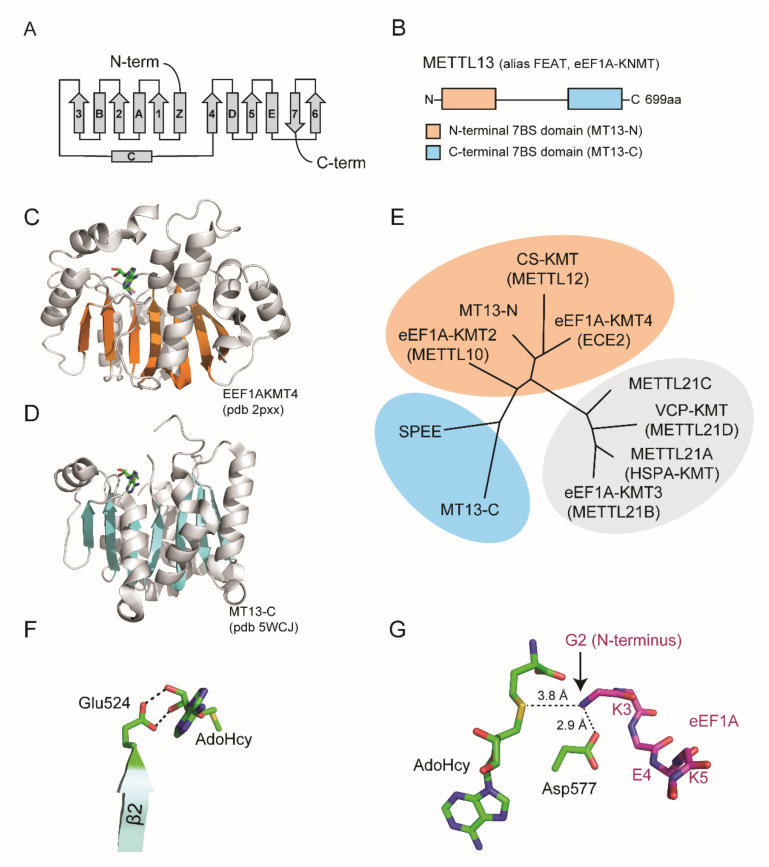
Domain organization and structure of METTL13. (**A**) Topology diagram of seven beta strand (7BS) methyltransferase fold. (**B**) Domain architecture of METTL13. (**C**,**D**) Structure of MT13-N-like eEF1A-KMT4 protein and the MT13-C domain. Ribbon representations are shown with beta strands highlighted in orange for (**C**) eEF1A-KMT4 (pdb # 2PXX) and blue for (**D**) MT13-C (pdb # 5WCJ). (**E**) Phylogenetic tree of METT13 domains and related methyltransferase enzymes. The tree was generated using the “Phylogeny.fr” platform [22] using METTL21A–D as an outgroup. (**F**) Structural model of MT13-C interaction with AdoHcy. Potential hydrogen bonds between Glu524 and the ribose moiety of AdoHcy are indicated (dashed lines). (**G**) Structural model of MT13-C and eEF1A N-terminal substrate peptide. Possible hydrogen bonds between METTL13-Asp577 and the eEF1A substrate peptide (stick representation, purple) are shown. The relative position and distance of the eEF1A N-terminus in relation to AdoHcy is indicated. The model was generated using the glide dock approach [23].

## 3. Biochemical Features

The catalytic activity of METTL13 is currently confined to the Nt and Lys55 of human eEF1A and the enzyme represents one out of five yet identified human eEF1A-KMTs (Figure 3). In vitro experiments with purified MT13-N and MT13-C domains have firmly demonstrated that MT13-N is responsible for dimethylation of Lys55 [10,11] and MT13-C is responsible for trimethylation of the protein Nt [11].

### 3.1. The METTL13 C-Terminal MT Domain

The Nt of proteins is most frequently acetylated [24]; only in rare occasions subject to methylation [25]. Notably, the major Nt acetyltransferase A complex (NatA) is reported to acetylate substrates with a small amino acid such as Gly, Ala or Ser in the second position, and after excision of the initiator Met residue [26]. Therefore, the discovery of eEF1A Nt methylation was somewhat unexpected. Nt methylation is also a rare PTM and MT13-C is to date one out of three validated human Nt MTs. Aside from MT13-C, Nt MT activity has been reported for the closely related NTMT1 (also called METTL11A and NRMT1) and NTMT2 (also called METTL11B and NRMT2) enzymes that target the second residue in proteins Met-(Ala/Pro/Ser)-Pro-Lys-, after iMet excision [27,28,29]. Recent studies have further refined the NTMT1/2 consensus motif to X-Pro-Lys/Arg (X = Gly, Ser, Pro, or Ala) [30,31,32].

The specificity of MT13-C has been explored in depth. First, In vitro MT assays have demonstrated that MT13-C primarily methylates as 50-55 kDa protein in METTL13 KO cells extracts, corresponding to the molecular weight of eEF1A [11]. Second, Protein MTs can recognize substrates in different ways. Conceptually, they may recognize a folded substrate, or a linear sequence motif present in the substrate. MTs targeting the flexible histone tails often belong to the latter class [33,34] whereas some 7BS-MTs such as VCP-KMT [35,36] and METTL21A [37,38] require a folded substrate. To explore the mode of substrate recognition for MT13-C and to identify potential additional substrates, a peptide array harboring systematic mutations of the eEF1A Nt was utilized to define a general recognition motif for MT13-C. These experiments indicated that the domain is capable of methylating peptide sequences corresponding to the eEF1A Nt and that it can methylate a linear motif corresponding to [GAP]-[KRFYQH]-E-[KRQHIL] (amino acid in eEF1A is underlined) in In vitro settings. This degenerate motif was used to identify ~50 candidate substrates in the human proteome. Notably, none of these candidate substrates were efficiently methylated by MT13-C, suggesting that the enzyme is highly specific for the eEF1A Nt [11].

The extent and spread of human eEF1A Nt methylation have been explored in a set of cells and tissues. The stoichiometry of methylation has been assessed through quantitative mass spectrometry experiments and revealed the site to be primarily trimethylated in mouse liver, kidney, and intestine [11] as well as in human HAP-1 [11], HEK-293 [21], and HeLa [5] cells.

Taken together, the collective body of biochemical experiments suggest that MT13-C is a highly specific MT for the eEF1A Nt, trimethylation is the predominant form and that it is widespread across mammalian cells and tissues.

### 3.2. The METTL13 N-Terminal MT Domain

The MT13-N domain has not been characterized to the same depth as MT13-C, mainly due to lower solubility levels of recombinant forms of the domain (unpublished observation by the author). Nonetheless, MT13-N has been firmly demonstrated to possess eEF1A-Lys55 activity in independent studies [10,11].

Dimethylation of eEF1A-Lys55 has been known for long [39] and in a recent methylproteomic study, we reported identification of the PTM in wide range of human cells including A549, HCT116, HEK293, HeLa, MCF7, and SY5Y as well as human tissue biopsies from liver, colon, and prostate [5]. The relative abundance of the different methylated forms of eEF1A-Lys55 has also been explored in a set of mammalian cells and tissues corresponding to RPE1, 293T, NCI-H2170, NCIH520, PaTu8902, T3M4, and U2OS cells [10] as well as mouse liver, kidney, and intestine [11]. In all analyzed cells and tissues, the dimethylated form of Lys55 has been predominant.

The link between METTL13 and methylation of eEF1A-Lys55 has been reported and validated in independent methylproteomics studies. Liu and colleagues in the Gozani lab showed that methylation of Lys55 was the only methylationsite strikingly under-represented in T3M4 METTL13 KO cells using a SILAC-based quantitative approach [10]. In similar experiments, we have reported both monomethylation of APOB-K1163 and dimethylation eEF1A-Lys55 as underrepresented in HAP-1 METTL13 KO cells [11]. Notably, the apparent significant under-representation of APOB-K1163me1 likely represents an experimental artefact and a remnant from bovine serum proteins added to the cell culture [40].

In summary, the body of data related to MT13-N suggests that the domain is highly specific for eEF1A-Lys55 and catalyzes dimethylation of the site in a broad range of mammalian cells and tissues.

## 4. Regulation

Proteins can be regulated at several different levels. Firstly, regulation can occur at the level of DNA and transcription. In addition, the stability of mRNA can be regulated. Finally, the function and stability of proteins can be regulated, for example by PTMs. Notably, METTL13 has been reported as regulated at all three levels.

At the level of transcription, HNL1 has been reported to upregulate METTL13 [41]. METTL13 protein levels are regulated at the mRNA level through the micro RNA miR-16, which targets the 3’ UTR of the METTL13 mRNA and mediates its degradation [42,43]. Notably, miR-16 has been linked to OvCa and the micro RNA is underrepresented in both ovarian cancer OvCa cell lines and primary ovarian tissues [44]. Specific implications of METTL13 and OvCa are further detailed below.

PTMs are key determinants of protein function and they can act as both positive and negative regulators of protein stability. For example, phosphorylation can both promote the stability and mediate degradation of proteins [45]. Moreover, distinct branches of poly-ubiquitination are linked to proteasomal degradation and autophagic clearance [46]. Intriguingly, METTL13 is reported to be both ubiquitinated and phosphorylated on multiple sites, whereof some are located in the active MT domains (Figure 4). However, the potential role of these PTMs in regulating the function and stability of METTL13 protein has not yet been explored.

In summary, little is known about the regulation of METTL13 but evidence suggests that it might be regulated at multiple levels including DNA, mRNA, and protein.

## 5. Cellular Features

Global transcriptomic data indicate that the METTL13 gene is ubiquitously expressed in human cells and tissues (Figure 5A). Moreover, recent exhaustive proteomics datasets also indicate that METTL13 protein is present in most, if not all, cells and tissues (Figure 5B). Notably, across multiple proteomics datasets, METTL13 is one of the more abundant METTL-proteins, and invariably the most abundant eEF1A-KMT (Figure 5B).

Human eEF1A represents the hitherto only validated METTL13 substrate and the molecular effects of METTL13 have mainly been linked to translation. Liu et al have shown that METTL13-mediated dimethylation of eEF1A-Lys55 increases the GTPase activity of the elongation factor and thereby increases the overall rate of translation and protein synthesis [10] (Figure 6). We have instead used a ribosome foot printing approach to assess the role of METTL13 mediated methylation in the context of translation dynamics. Our method relies on the well-established notion that the ribosome samples cellular eEF1A-GTP-aminoacyl-tRNA complexes in its acceptor (A)-site. Consequently, the frequency of a codon in the A-site can be used as a proxy for its rate of translation [49]. The analysis revealed that cells lacking METTL13-mediated eEF1A methylation displayed a faster translation of histidine codons and a slower translation of alanine codons [11] (Figure 6).

Notably, the deletion of other human eEF1A-KMTs [18,50] as well as MTs targeting ribosomal proteins [51] and rRNA [52] have led to related phenotypes with altered translation of specific codons. Collectively, these observations corroborate that methylation of the translational apparatus is frequent and represents a mechanism to regulate proteins synthesis [7,53]. In analogy to the “histone code” in epigenetics [54,55], it has been proposed that the multiple reported PTMs on eEF1A may constitute an “eEF1A code” [56,57] that dynamically regulates gene activity at the level of translation.

## 6. Connection to Diseases

METTL13 has been evaluated in the context of disease biology, mainly cancer, but the reports are somewhat scattered and do not point in a unified direction. This can be a consequence of cell-type specific functions of METTL13 or pleiotropic phenotypes related to its general role as a regulator of protein synthesis.

A seminal study by Takahashi et al first linked METTL13 to cancer [58]. They generated a mouse model that over-expressed METTL13 in a tissue-specific manner and observed that mice developed tumors in the organs where the gene was overexpressed, suggesting the enzyme is a general driver of tumorigenesis. In subsequent studies from the Takahashi lab, METTL13 plasma levels were reported as elevated in several cancer types, and specifically in OvCa [59], suggesting the protein may have clinical utility as a biomarker.

The abundance of METTL13 has been linked to both favorable and poor cancer prognosis. For clear cell renal cell carcinoma, high levels of METTL13 has been linked to favorable prognosis and the enzyme has been reported to inhibit growth and metastasis [60]. In bladder cancer, METTL13 has been reported to negatively regulate key cancer hallmarks including proliferation, migration and invasion [61].

In contrast, METTL13 levels have been reported as elevated and linked to unfavorable prognosis for other cancer types. In head and neck squamous cell carcinoma, METTL13 expression is increased at both the transcript and protein level and high levels are associated with poor prognosis [62]. In hepatocellular carcinoma, METTL13 has been implied in mediating tumor growth and metastasis [41]. Finally, a recent study revealed that both METTL13 and eEF1A-K55me2 levels are upregulated in pancreatic and lung cancer and high levels of both these markers were linked to low patient survival [10]. The authors also convincingly demonstrated that METTL13 depletion sensitized cancer cells to PI3K and mTOR pathway inhibition [10]. Importantly, independent studies have shown that downregulation of METTL13 levels by miR-16 induces apoptosis [42]. Taken together, this indicates that inhibition of METTL13 may represent an effective strategy in combinatorial cancer therapy approaches.

METTL13 has 157 mutations annotated in the Catalog of Somatic Mutations In Cancer (COSMIC) database (Figure 7) and it has been highlighted as the most mutated METTL protein in comprehensive transcriptomics cancer datasets [63]. However, as METTL13 is a dual 7BS domain MT it is also one of the larger 7BS MTs which can represent an explanation for the high number of annotated mutations.

In addition to the numerous links to different types of cancer, METTL13 has also been associated with hearing loss. In detail, a dominant mutation corresponding to Arg544Gln in the METTL13 protein has been linked to deafness [65].

In summary, aberrant expression of METTL13 has been linked to a wide range of cancers and it has been suggested to function as both oncogene and tumor suppressor.

## 7. Conclusions and Outlook

During the last decade, significant discoveries have been made to increase the understanding of METTL13 enzymatic activity, cellular features, and links to disease. Future research efforts will likely extend on the current knowledge status, especially how aberrantly regulated METTL13 relates to cancer etiology and progression.

Biochemically, future focus will likely be devoted to comprehending the potential dynamic nature of METTL13-mediated eEF1A methylation. While methylation of histone proteins [66] and eEF1A-Lys165 [50] have been reported as dynamic, there are yet no reports of potential demethylases targeting the eEF1A Nt or Lys55. Here a combination of heavy methyl [67] and dynamic SILAC [68] can be used to assess cellular turnover of both bulk and methylation modified species of eEF1A, to uncover potential methylation dynamics.

From a clinical perspective, METTL13 is clearly linked to key cancer hallmarks and recent evidence suggests that combined targeting of METTL13 and key cellular signaling pathways may represent an effective therapeutic strategy in cancer management. Here, large-scale synthetic lethality studies using genome-wide CRISPR KO libraries can globally uncover co-dependencies of METTL13 and other genes, particularly key signaling hubs. Such experiments have potential to uncover novel strategies for combination cancer therapy.

## Figures and Tables

**Figure 1 life-11-01121-f001:**
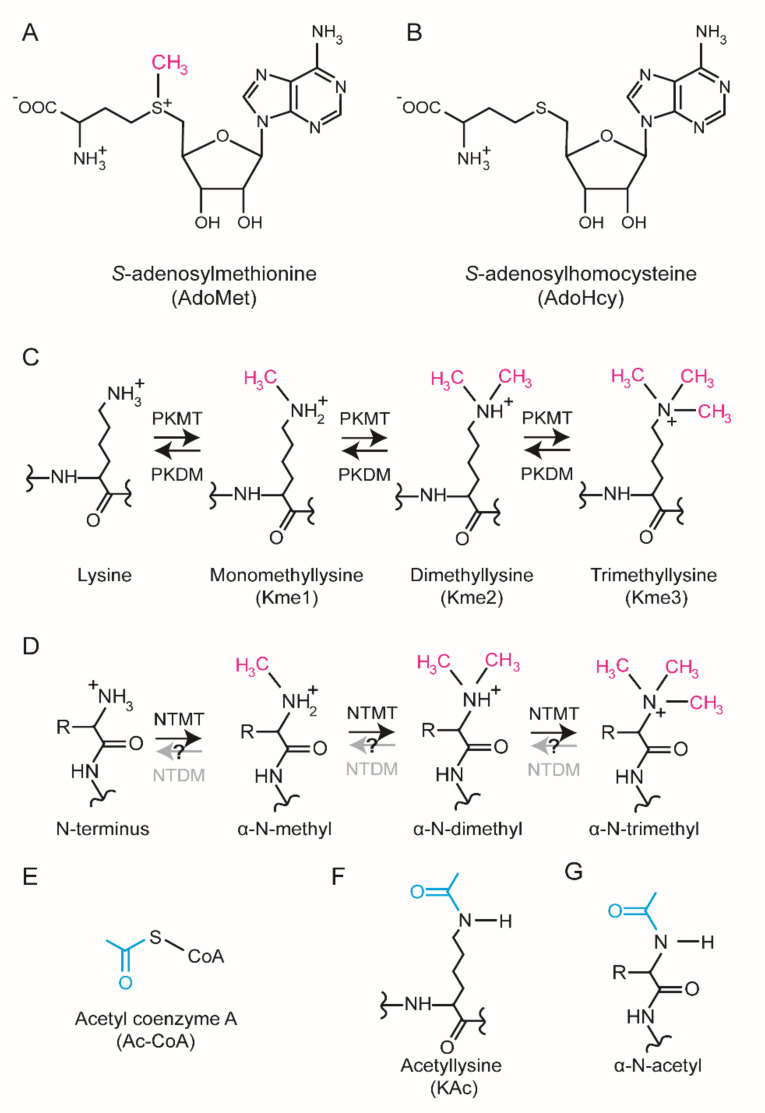
Biochemistry of protein lysine and N-terminal methylation. (**A**,**B**) Structures of AdoMet and AdoHcy. The chemical structures of the methyl donor (**A**) *S*-adenosylmethionine (AdoMet) and (**B**) the demethylated counterpart (AdoHcy) are shown. The transferred methyl group is highlighted (magenta). (**C**) Biochemistry of lysine methylation. Consecutive protein lysine methyltransferase (PKMT)-mediated methylation can introduce up to three methyl groups in a lysine side chain. The methyl groups can be enzymatically removed by protein lysine demethylase (PKDM) enzymes. (**D**) Biochemistry of protein N-terminal methylation. Consecutive protein N-terminal methyltransferase (NTMT)-mediated methylation can introduce up to three methyl groups on the α-amino group of proteins. There is yet no evidence of protein N-terminal demethylase (NTDM) enzymes, but their potential enzymatic activity is indicated (grey arrow, question mark). (**E**) Structure of acetyl-coenzyme A. The transferred acetyl group is highlighted (cyan). (**F**,**G**) Structures of (**F**) acetyl lysine and (**G**) α-N-acetylated protein terminus are shown.

**Figure 3 life-11-01121-f003:**
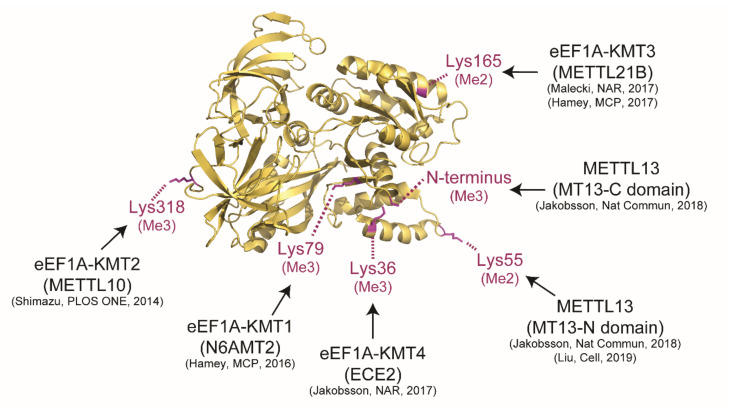
Methylation of human eukaryotic elongation factor I alpha. The structure of eEF1A (pdb # 1F60) is shown in ribbon representation (gold). Key methylation sites in human eEF1A with the predominant methylated forms (magenta) as well as the responsible MT enzymes are indicated.

**Figure 4 life-11-01121-f004:**
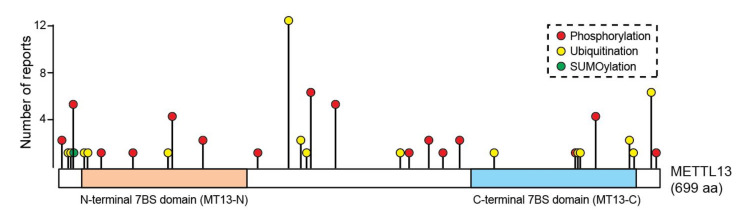
The PTM landscape of METTL13. The data were retrieved from the “PhosphoSitePlus” database [47] (www.phosphosite.org; (accessed on 15 September 2021)).

**Figure 5 life-11-01121-f005:**
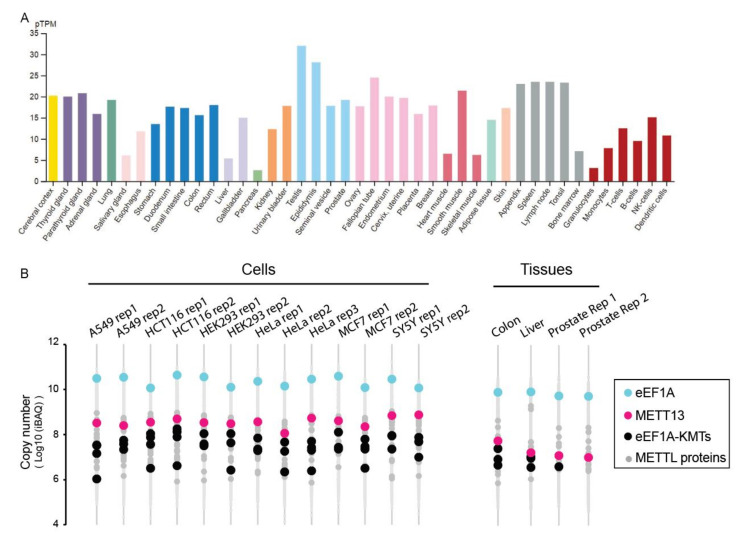
METTL13 expression in cells and tissues. (**A**) RNA data showing protein-transcripts per million in a range of human organs and cell types. The data were retrieved from the human protein atlas (HPA, https://www.proteinatlas.org/ (accessed on 15 September 2021)) and color coded according to tissue or cell type. (**B**) Proteome data showing METTL13 protein levels in cultured human cells and primary tissue biopsies. The data were retrieved from ProteomeXchange (dataset PXD004452) [48]. eEF1A and METTL13 are highlighted and other eEF1A-KMTs (METTL10, EEF1AKMT4, N6AMT2, METTL21B) and METTL- proteins are indicated.

**Figure 6 life-11-01121-f006:**
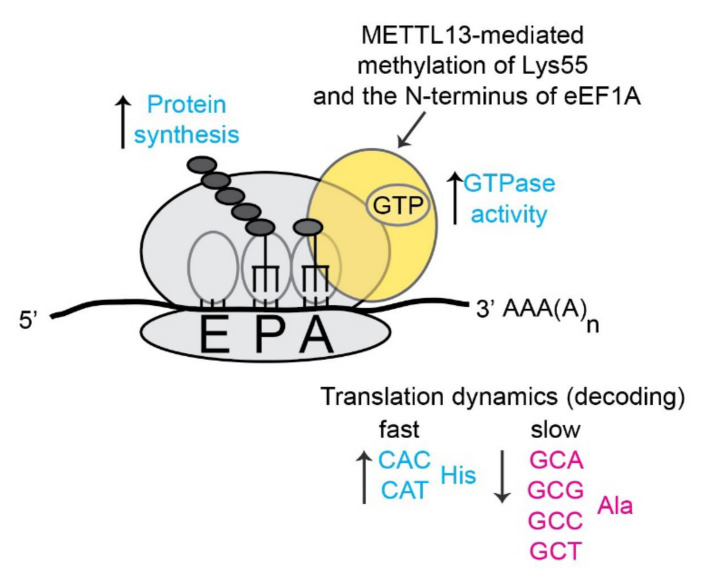
Molecular effects of METTL13-mediated eEF1A methylation on translation.

**Figure 7 life-11-01121-f007:**
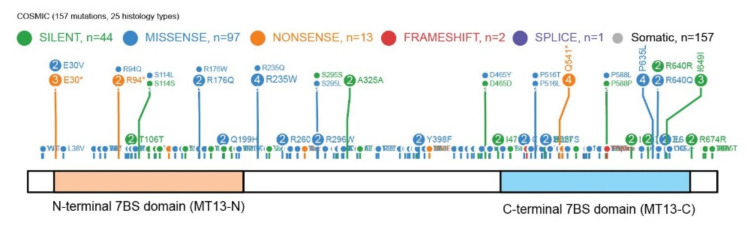
The mutational landscape of METTL13. The data was retrieved from the COSMIC database and the visualization is modified from ProteinPaint [64].

## Data Availability

The proteomics data used to generate Figure 5B is available through ProteomeXchange (dataset PXD004452).

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
