# Peer review of "Structure, Activity and Function of the Dual Protein Lysine and Protein N-Terminal Methyltransferase METTL13"

_life, 2021, doi:10.3390/life11111121_

Round 1
Reviewer 1 Report
This is a timely and relatively comprehensive review on the METTL13 protein that has recently gained considerable interest due to its identification as a dual protein lysine and protein N-terminal methyltransferase, targeting the elongation factor eEF1A. The article was prepared by dr. Jakobsson, a well-experienced researcher in the biochemistry of METTL13, who is indeed the right person for writing such a work, ensuring its high scientific quality.
I found this review easy to read, well-structured, and documented. The structural, biochemical, and cellular aspects of METTL13 were covered with a good style and with good use of available literature. The text was also illustrated with seven well-prepared and informative figures. Some issues, particularly the regulation of activity, were only briefly discussed, but this is entirely understandable given a complete lack of literature on the problems. Some minor corrections of typos such as “e” (p. 4, line 74) or “well-stablished” (p. 9, line 221) are also required, but this might as well be done at the proof stage.
To sum up, this is a timely review on an interesting topic. No articles covering similar subjects are currently available, and I am sure that this work will attract considerable interest from the community.
Reviewer 2 Report
In this manuscript, Jakobsson M.E. summarized the structure, activity, and function of METTL13, which is a dual protein lysine and N-terminal methyltransferase. The manuscript is outlined clearly. However, the scope of this review has a narrow scope as there are only ~20 references on METTL13/FEAT. Among them, one of the major references is from the author. Instead of summarizing the findings of METTL13, not many new insights or discussions were offered in this review, which did not expand our knowledge on METTL13. Meanwhile, there are several errors listed below:
- Structures in Figures 1A and B are not correct. The carboxylic acid group was missed in both structures. The structure in Figure 1E should remove the carbon atom between S and CoA.
- Page 6 line 126, the latter “NTMT1” should be NTMT2.
- Line 128, new literature on structural studies indicates the motif is X-P-K/R.
Round 2
Reviewer 2 Report
Despite the slight improvement on part 7, more in-depth and thorough discussions are needed to add value of this review.
Line 47-49: Lysine is charged under the physiological condition. So the second point is not correct as lysine methylation does not change the charge state.
Original references on substrate recognition of NTMT1/2 should be included.
- https://pubmed.ncbi.nlm.nih.gov/26543161/
- https://www.nature.com/articles/s42003-018-0196-2
